# Patient-specific Biomolecular Instruction tuning of Graph-LLMs

## Abstract

Proteomics data is essential to pathogenic understanding of a disease phenotype. In cancer, analysis of molecular signatures enables precision medicine through the identification of biological processes that drive individualized tumor progression, therapeutic resistance, and clinical heterogeneity. Recent advances in multimodal large language models (LLMs) have shown remarkable capacity to integrate and reason across heterogeneous data modalities. However, performing multi-modal language modeling for molecular understanding of patient-specific proteomics remains a significant challenge due to two barriers: (1) the lack of instruction-tuning datasets that enable clinical interpretation from proteomics data, and (2) the absence of language modeling architectures designed to capture the rich heterogeneity of molecular data. In this work, we introduce CPTAC-PROTSTRUCT, the first instruction tuning dataset for proteomic understanding of oncology, comprising over 370k open-ended examples derived from more than 1000 patients curated from the largest United States proteomics cancer study (CPTAC). Additionally, we propose KRONOS (Knowledge Representation of patient Omics Networks in Oncology via Structured tuning), a novel graph-LLM framework that leverages molecular interaction topology with proteomics to learn patient-specific graph representations for enhanced clinical reasoning. We show that KRONOS achieves consistent improvements across benchmark clinical tasks, with AUC performance of up to $0.857 \pm 0.025$ in prognostic tasks such as mortality prediction, cancer type OS prediction, and tumor stage classification from proteomics data. Ultimately, this approach empowers LLMs to understand patient-level pathogenesis, advancing precision medicine through more accurate diagnosis, prognosis, and treatment stratification.

## 1 Introduction

Cancer represents one of the most complex and heterogeneous diseases known to biomedicine, where genomic mutations alone fail to explain the complex phenotypic diversity, treatment patterns, and clinically observed patient outcomes (Gerlinger & Swanton, 2013). However, the exponential growth of high-throughput proteomics data has enabled opportunities to capture the molecular landscape driving cancer pathogenesis, enabling scientists to understand sophisticated disease mechanisms and therapeutic targets (Li et al., 2024b; Savage et al., 2024; Chen et al., 2023). Unlike the static nature of molecular genomics (aside from additional mutations), proteomics is an immediate manifestation of a patient's disease pathogenesis by reflecting individual, real-time cellular responses to pathological processes, environmental stimuli, and therapeutic interventions (Al-Amrani et al., 2021; Guo et al., 2023). Despite being rich in biological information, proteomics is highly variable, and understanding how these molecular signals contribute to a patient outcome requires advanced approaches that can identify hidden patterns within complex molecular datasets and enable personalized treatment strategies.

Traditional proteomics analysis have largely focused on individual protein abundance changes, often overlooking the interactive interplay between molecules, and the implications of these interactions (Krogan et al., 2011). However, recent advances in graph representation learning and identification of validated protein interactions in biological literature have allowed scientists to ground deep learning with biological context, through structure-aware graph neural networks that integrate protein-protein interactions with patient-specific proteomics signatures (Heim et al., 2022; Yuan

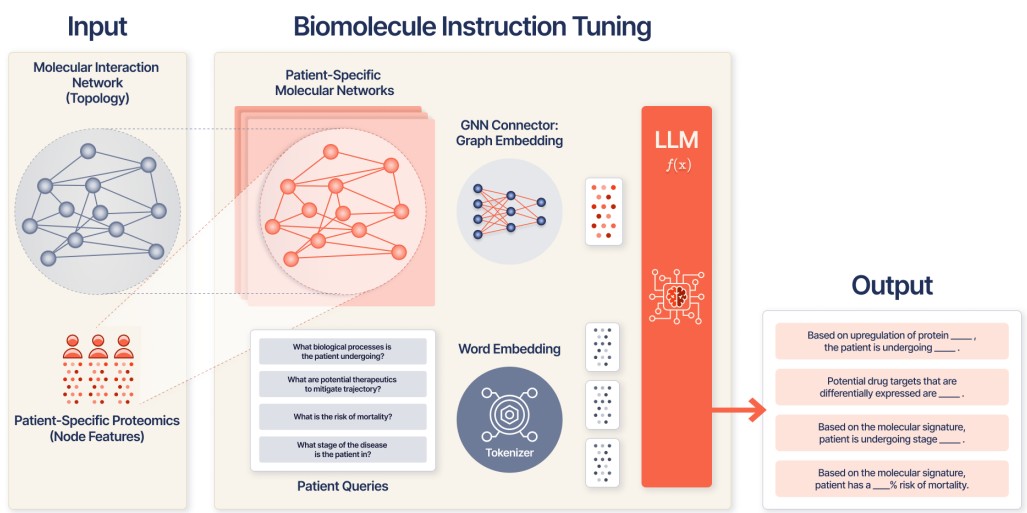

Figure 1: Model architecture of KRONOS.

et al., 2022; Li et al., 2022a). Additionally, the rise of LLMs in the clinical domain and instruction tuning (Liu et al., 2023) paradigms have allowed multi-modal reasoning grounded in free text, enabling integration and biomedical reasoning of diverse data types including radiology and pathology images (Kather et al., 2024; Sun et al., 2024), patient EHR data (Wang et al., 2025), clinical knowledge (Singhal et al., 2023), and therapeutics (Huang et al., 2024). However, there still remains a significant challenge in establishing a multi-modal large language model to reason on individualized proteomics data to interpret intricate biological interactions and associated clinical outcomes.

More specifically, there are critical limitations in current literature that prevent individualized semantic molecular reasoning: (1) existing patient-level instruction-tuning datasets focus on general clinical tasks and lack the molecular specificity needed for proteomics interpretation, creating a training data gap between protein-level measurements and prognostic reasoning, and (2) while large language models excel at reasoning over textual data, they lack native capabilities to process and interpret the complex biomolecular interactions inherent to proteomics data. These limitations underscore the need for a unified architecture that can seamlessly accomodate graph-structured protein interaction data with patient-specific molecular signatures, while enabling natural language reasoning about complex biological relationships and clinical outcomes.

To address these limitations, we introduce CPTAC-PROTSTRUCT (Section 3), the first patient-level instruction tuning dataset for molecular oncology understanding, comprising over 380,000 examples that bridge individualized proteomic profiles with clinical reasoning tasks from CPTAC (National Cancer Institute, 2023). Furthermore, we propose KRONOS (Knowledge Representation of patient Omics Networks in Oncology via Structured LLM tuning), a unified graph-LLM framework that integrates molecular interaction topology with patient-specific proteomics data for prognostic assessment through graph representation learning within the language modeling architecture. Through our experiments, KRONOS (Section 4) achieves competitive performance across several prognostic usecases, advancing precision medicine through more accurate patient stratification from individualized proteomics signatures.

## 2 RELATED WORK

### 2.1 MOLECULAR INTERACTION AWARE GRAPH DEEP LEARNING IN OMICS

Graph-based approaches have emerged as powerful tools for modeling complex biological relationships in omics data, with protein-protein interaction (PPI) networks serving as fundamental

| | |
|---|---|
| Schema Alignment Questions | Find the proteins whose measurements exceed two standard deviations from the mean value. |
| | How does the amount of RPL35 compare to {protein} in terms of relative abundance? |
| | Could you tell me the concentration of {protein} in this patient? |
| | Report the expression level of {protein}. |
| | Which proteins belong to the uppermost 90% when ranked by their abundance? |
| Clinical Reasoning Questions | What does the molecular network predict for treatment response? |
| | Based on the protein expression network, predict the tumor code. |
| | Predict overall survival days based on the molecular profile. |
| | Determine histologic grade and pathological stage from the molecular network. |
| | Analyze recurrence risk using the patient's molecular signature data. |

Table 1: Examples of schema alignment and clinical reasoning questions.

structural scaffolds for understanding molecular mechanisms. The STRING database has provided experimentally-validated protein-protein interaction networks across thousands of organisms (Szklarczyk et al., 2019). Building on such resources, several methods have demonstrated the effectiveness of integrating molecular data with graph neural networks on PPI networks. EMOGI pioneered explainable graph convolutional networks for cancer gene prediction by combining pan-cancer multiomics data with PPI networks (Schulte-Sasse et al., 2021), while spectral-based convolutional approaches have successfully integrated proteomics and transcriptomics data for complex disease classification (Zhuang et al., 2023). GNN-SubNet advanced explainable disease subnetwork detection using PPI topology with multi-omics node features (Pfeifer et al., 2022), and MTGCL introduced multi-task graph contrastive learning to address supervised signal sparsity in cancer driver gene identification (Zhou et al., 2025; Li et al., 2022b). More recently, CGMega developed explainable graph attention frameworks for cancer gene module dissection (Li et al., 2024a), while TREE extended this paradigm using transformer-based models across multiple biological interaction networks (Su et al., 2025). These methods collectively demonstrate that leveraging explicit structural relationships in PPI networks provides biologically meaningful priors that significantly enhance both performance and interpretability compared to traditional approaches. Building upon this foundation, our work extends to the proteomics domain by developing the first individualized PPI-graph LLM that combines patient-specific protein expression and string PPI network topology to enable semantic alignment of prognostic outcomes.

## 2.2 CLINICAL MULTI-MODAL INSTRUCTION TUNING

Instruction tuning has emerged as a powerful approach for developing specialized AI assistants capable of processing complex biological and clinical data. MIMIC-Instr pioneered large-scale instruction tuning for electronic health records with over 400K instruction-following examples, enabling LLMs to process complex EHR structures (Wang et al., 2024). In protein analysis, structure-enhanced protein instruction tuning has demonstrated the potential for general-purpose protein understanding by combining sequence and structural information in LLM training (Wu et al., 2025). Multimodal approaches include LLaVA-Med, which achieved efficient biomedical vision-language instruction tuning using PubMed figure-caption pairs and GPT-4 generated instruction data (Li et al., 2023), and MEIT, which introduced ECG instruction tuning frameworks aligning cardiac signals with clinical reports (Liu et al., 2024). Recent advances include Me-LLaMA, combining continual pretraining with instruction tuning using 129 billion biomedical tokens (Chen et al., 2025), Dr-LLaVA incorporating symbolic clinical grounding for diagnostic conversations (Goldgof et al., 2024), and BioMistral-NLU demonstrating improved generalizability across medical natural language understanding tasks (Yang et al., 2024). These methods collectively establish instruction tuning as an effective technique for adapting foundation models to specialized biological applications (Butte et al., 2024).

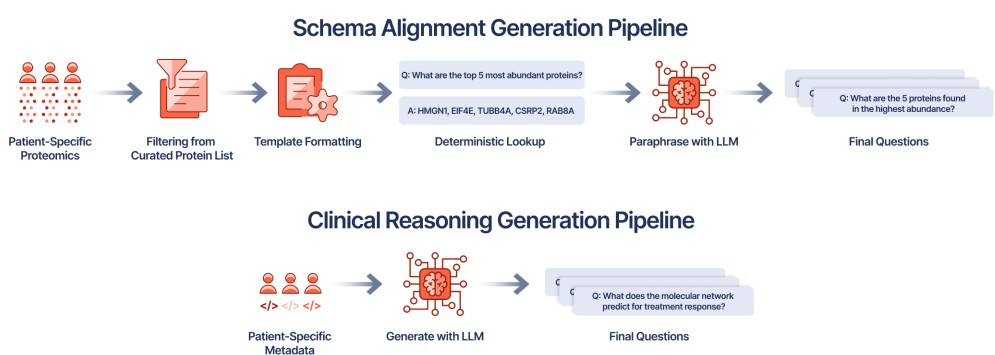

Figure 2: CPTAC-PROTSTRUCT instruction generation pipeline.

## 3 PROTEOMICS INSTRUCTION TUNING

Advanced technologies have been developed to learn optimal representations of individual molecular data. However, semantic reasoning on individualized proteomics data has still been a challenging task, primarily due to the biological expertise needed to curate instruction datasets that bridge the gap between complex proteomic profiles and clinically meaningful outcomes. Thus, the development of specialized instruction-tuning datasets that enable language models to perform sophisticated molecular reasoning and generate accurate diagnostic insights from patient-specific proteomics data is imperative for LLM understanding of complex biological systems.

To enable a general-purpose LLM to comprehend molecular insights, we first train it to navigate the proteomics modality space through specialized schema alignment. Following this initial adaptation, structured fine-tuning is required to leverage this new modality for generating clinical reasoning and inferring patient outcomes. Drawing from the demonstrated efficacy of utilizing large-scale LLMs to generate instruction-following data (Liu et al., 2023), we created CPTAC-PROTSTRUCT, the first proteomics instruction-following dataset derived from individual protemics profiles for clinical outcomes. CPTAC-PROTSTRUCT includes 2 subsets: a schema alignment instruction dataset and prognostic reasoning instruction dataset, which are used in different training stages. An overview of the creation of both subsets of instruction pairs can be found in Figure 2.

### 3.1 CPTAC PROTEOMICS DATABASE PREPARATION

We construct our cohort from the dataset from the Clinical Proteomic Tumor Analysis Consortium (CPTAC) study (National Cancer Institute, 2023), utilizing the extraction pipeline outlined in (Payne et al., 2021). Each patient was represented by at least one sample in the dataset, with at most of one tumor sample and one normal sample per patient. To maintain data quality, we applied patient-level filtering by removing subjects with more than 35% missing protein data, resulting in a 5% dropout rate. We then identified a core set of proteins by selecting those present in more than 70% of both tumor and normal samples, reducing the feature space from 16,869 to 5,078 proteins. Missing values in the core protein set were imputed using Multiple Imputation by Chained Equations (MICE) (Van Buuren & Groothuis-Oudshoorn, 2011) within each cancer type to preserve tissue-specific patterns. Finally, we performed variance-based feature selection by removing the bottom 5% lowest variance proteins per dataset to eliminate uninformative features, yielding our final preprocessed dataset of 1,658 samples (1,086 tumor, 572 normal samples) across 4,824 proteins spanning 10 cancer types.

### 3.2 PROTEIN CURATION FOR INSTRUCTION DATA

Due to the sheer volume of proteins in the processed CPTAC dataset, we recognize that creating an instruction tuning dataset that queries every single protein would be computationally prohibitive and dilute learning with low-impact features. Thus, we limited our dataset generation to only query high-

| Schema Alignment | |
|---|---|
| **User** | Indicate the relative abundance of SMARCA5. |
| **Gold** | The abundance level of SMARCA5 in this patient is 0.101. |
| **Vicuna** | I'm sorry, but I don't have information on the relative abundance of SMARCA5. Can you provide more context or clarify your question? |
| **KRONOS** | SMARCA5 has an abundance of 0.101. |

| Clinical Reasoning | |
|---|---|
| **User** | Based on the identified molecular profile, will there be an increased likelihood of cancer recurrence in this patient? |
| **Gold** | The patient is predicted to experience recurrence. |
| **Vicuna** | As an AI language model, I cannot provide a diagnosis or prognosis for a specific patient, as I do not have access to their medical history or current health status. |
| **KRONOS** | The patient is at predicted risk for recurrence. |
| **User** | Based on the protein abundance profile, which histologic subtype is indicated by this molecular signature? |
| **Gold** | Based on the molecular signature, the histologic type is Glioblastoma. |
| **Vicuna** | The protein abundance profile in the breast cancer cohort included 23 proteins, including HER2, ERBB2, and PR. The most abundant protein in the tumor samples was HER2, which was significantly more abundant in TNBCs compared to LCIS and HOC cases. The other 22 proteins were relatively equally abundant across the three subtypes. |
| **KRONOS** | The patient has glioblastoma. |

Table 2: Model generation comparison of schema alignment and clinical reasoning.

impact proteins that are most clinically relevant and biologically informative for cancer diagnosis and prognosis.

We employed a multi-faceted approach to identify these critical proteins through three complementary categories: high-impact proteins, high variance proteins, and cancer associated proteins. We considered high-impact proteins to be markers that are functionally critical across many biological processes. These were identified through PPI network centrality analysis using degree centrality with a top 10% rank threshold to capture proteins that significantly influence biological interaction topology. We further incorporated pathway analysis using Reactome (Jassal et al., 2020) to prioritize proteins involved in fundamental processes, including core cell cycle regulation, DNA damage response, metabolism, and established drug targets. Furthermore, we identified high variance proteins by selecting proteins with high variability across all samples using a 10% threshold. Finally, we extracted cancer-associated proteins which are specifically implicated in oncogenesis, tumor progression, or therapeutic response, from two authoritative databases: OncoKB (Chakravarty et al., 2017), which provides annotations of oncogenes, and COSMIC (Tate et al., 2019), a catalog of cancer somatic mutations.

This curated list of proteins represents clinically actionable and biologically informative features while maintaining computational tractability for comprehensive instruction dataset generation.

## 3.3 CPTAC-PROTSTRUCT: SCHEMA ALIGNMENT GENERATION

To generate optimal instruction-following questions to navigate the proteomics modality space, we generated a schema alignment subset designed to enable associations between patient-specific protein abundance values with their corresponding semantic representations. We developed five question types to comprehensively cover proteomics data interpretation: (1) direct protein abundance queries to request specific abundance values, (2) abundance threshold queries that ask about proteins within a certain threshold, (3) ranking and ordering queries that sort proteins by abundance levels, (4) comparative abundance queries that compare expression between multiple proteins, and (5) interaction network-based abundance queries that explore protein relationships within interaction networks. To ensure linguistic diversity and preserve natural language patterns, all questions were

paraphrased using DeepSeek-R1-Distill-Qwen-32B (DeepSeek Team, 2024), resulting in 354,812 final schema alignment questions with varied linguistic expressions while maintaining semantic consistency. Examples of schema alignment questions are provided in Table 1.

### 3.4 CPTAC-PROTSTRUCT: CLINICAL REASONING GENERATION

Expectations for molecular oncology AI often go beyond protein abundance queries to performing diagnostic and prognostic reasoning with proteomics data. To align model training with this goal, we created diverse instruction-following data focused on patient-centric clinical reasoning using DeepSeek-R1-Distill-Qwen-32B. Specifically, we prompted it to generate QA pairs that resemble those oncologists might ask when interpreting patient proteomic profiles in clinical settings. We manually created few-shot examples in the prompt to demonstrate how to generate high-quality QA pairs, and leveraged associated clinical metadata as contextual input. Compared to raw protein expression values alone, this clinical metadata provides essential prognostic context that makes the generated questions more suitable for clinical reasoning. In this way, we generated approximately 26,157 clinical reasoning QA pairs to equip the model with the ability to make meaningful interpretations of proteomic data. Note that while clinical metadata enhances instruction-tuning quality, our foundation model inputs consist primarily of the proteomic abundance profiles themselves, ensuring the model learns to extract clinical insights directly from molecular data. Examples of clinical reasoning questions are provided in Table 1

## 4 KRONOS: KNOWLEDGE REPRESENTATION OF PATIENT OMICS NETWORKS IN ONCOLOGY VIA STRUCTURED TUNING

With the finalized instruction tuning pairs and their corresponding patient-specific molecular signatures, we introduce KRONOS (Knowledge Representation of patient Omics Networks in Oncology via Structured tuning ), a novel graph-LLM architecture depicted in Figure 1 that processes individualized proteomic profiles and generates biologically contextualized representations through integration with protein-protein interaction networks. First, patient-specific proteomics data is embedded as node features within the corresponding protein nodes of the STRING PPI network Szklarczyk et al. (2019), resulting in a molecular network for every patient. These proteomics-informed molecular graphs are subsequently processed through a graph neural network, with the corresponding graph representation integrated as a specialized token into a generalized LLM for downstream instruction tuning. This pipeline enables LLM reasoning over structured biological interactions, allowing the model to leverage both molecular-level mechanistic insights and patient-specific expression patterns for clinical predictions.

### 4.1 PROBLEM SETUP

Let $\mathcal{D} = (P_i, q_i, a_i)_N$ be our instruction tuning dataset, where $P_i \in \mathbb{R}^{M \times d}$, and $N$ denotes the number of triplets where each patient's protein expression data is paired with instruction-answer pairs. Each patient $i$ is associated with a personalized protein–protein interaction network $\mathcal{G}_i = (\mathcal{V}_i, \mathcal{E}_i, X_i)$, where $X_i \in \mathbb{R}^{|\mathcal{V}_i| \times d}$ represents proteomics-informed node features. These personalized molecular graphs integrate STRING-derived interaction topology with proteomics data, enabling patient-specific modeling of molecular mechanisms.

To create a representation for the entire molecular interaction graph, we apply a graph neural network (GNN) encoder $\phi_{\text{PPI}}$ to each patient-specific PPI graph $\mathcal{G}_i = (\mathcal{V}_i, \mathcal{E}_i, X_i)$, where $\mathcal{V}_i$ and $\mathcal{E}_i$ denote the set of protein nodes and interactions, respectively, and $X_i$ contains omics-informed node features. The GNN encoder computes hidden node representations through $L$ layers of message passing, starting from initial node features $\mathbf{h}_v^{(0)} = \mathbf{x}_v$, where $\mathbf{x}_v$ is the omics feature vector for protein node $v$. At each layer $\ell = 1, \ldots, L$, the hidden representation of node $v \in \mathcal{V}_i$ is updated as:

$$\mathbf{h}_v^{(\ell)} = \sigma\left(\mathbf{W}^{(\ell)} \cdot \text{AGGREGATE}^{(\ell)}\left(\left\{\mathbf{h}_u^{(\ell-1)} : u \in \mathcal{N}(v) \cup \{v\}\right\}\right)\right), \quad (1)$$

where $\mathcal{N}(v)$ denotes the set of neighbors of $v$, $\mathbf{W}^{(\ell)}$ is a trainable weight matrix, $\sigma$ is a non-linear activation function (e.g., ReLU), and AGGREGATE$^{(\ell)}$ is a permutation-invariant function such as

mean, sum, or attention. After the final layer, we obtain the set of node representations $\{\mathbf{h}_v^{(L)}\}_{v \in \mathcal{V}_i}$, which are aggregated using a READOUT function (e.g., max pooling) to produce a graph-level embedding

$$\mathbf{z}_i = \phi_{\text{PPI}}(\mathcal{G}_i) = \text{READOUT}\left(\left\{\mathbf{h}_v^{(L)} : v \in \mathcal{V}_i\right\}\right). \tag{2}$$

To align the molecular graph representation with the LLM's embedding space, we employ a dense connector network

$$\mathbf{e}_i = \phi_{\text{connector}}(\mathbf{z}_i) \tag{3}$$

where $\mathbf{W}_{\text{dense}} \in \mathbb{R}^{d_{\text{llm}} \times d_{\text{graph}}}$ and $\mathbf{b} \in \mathbb{R}^{d_{\text{llm}}}$. The output $\mathbf{e}_i$ matches the LLM token embedding dimension. The processed molecular embedding $\mathbf{e}_i$ is integrated into the instruction as a special token. Let $\mathbf{T}_{\text{text}} = [\mathbf{t}_1, \ldots, \mathbf{t}_n]$ be token embeddings of $q_i$. The multi-modal input is

$$\mathbf{T}_{\text{multi}} = [\mathbf{e}_i, \mathbf{t}_1, \ldots, \mathbf{t}_n], \tag{4}$$

which is processed by the LLM as

$$\mathbf{H} = \text{LLM}(\mathbf{T}_{\text{multi}}). \tag{5}$$

### 4.2 TRAINING WITH CURRICULUM LEARNING

Inspired by LLaVA (Liu et al., 2023), we use a two-stage training approach, first bridging the gap between general text and proteomics data, then developing molecular reasoning capabilities for prognostic interpretation.

#### 4.2.1 STAGE 1: TRAINING FOR SCHEMA ALIGNMENT

We employ the paraphrased 354,812 template-generated QA pairs for stage 1 training. For each patient, given the PPI graph and proteomics instruction, we train the model to generate appropriate responses. We freeze only the LLM backbone, updating both the connector network and the graph encoder. This allows training of a representation space that directly aligns with the sematic space of the LLM, and enables the LLM to interpret molecular graph representations to bridge the modality gap between general text and proteomics data. Hyperparameter search spaces are stated in the Appendix.

#### 4.2.2 STAGE 2: TRAINING FOR CLINICAL REASONING

In this stage, we fine-tune the model for complex instruction following and molecular reasoning. We utilize the remaining 26,157 QA pairs for proteomics reasoning tasks, updating both the LLM, connector, and the GNN encoder. This enables the model to perform advanced molecular reasoning beyond simple information extraction. Hyperparameter search spaces are stated in the Appendix.

## 5 EXPERIMENTS

### 5.1 PERFORMANCE ON STANDARD CLINICAL PREDICTIVE BENCHMARKS

To evaluate KRONOS on the CPTAC/TCGA dataset, we identify 4 critical outcomes for patient prognosis: mortality prediction (patient survival status), cancer type classification, overall survival estimation, and disease stage prediction. We compare KRONOS against four baseline categories: linear modeling approaches (Lasso, ElasticNet, SVC), classical deep learning methods (3-layer and 5-layer MLPs), patient similarity network node classification approaches (Tate et al., 2019; Shreykar et al., 2018), and biomolecular graph classification approaches (Zitnik et al., 2018; Jha et al., 2023).

For similarity network node classification and PPI-graph classification models, training paradigms and network creation are set identical to recent literature (Wang et al., 2021; Schulte-Sasse et al., 2021; Zhuang et al., 2023; Pfeifer et al., 2022; Li et al., 2024a), and trained with various graph neural networks, GAT (Veličković et al., 2018), GraphSage Hamilton et al. (2017), and GINConv Xu et al. (2019), for optimal performance. All models were evaluated using a 5-fold nested cross validation identical grid search parameters. All hyperparameters are explained in the supplementary table 1. The LLM used for all experiments is Vicuna7bv1.5 Chiang et al. (2023), as recent works in instruction-tuning literature all adopt this model, for fair comparison and an established baseline performance in biomedical domain adaptation tasks.

| Model | Mortality Pred. | | Cancer Type | | OS Prediction | | Stage Class. | |
|---|---|---|---|---|---|---|---|---|
| | AUC | F1 | AUC | Macro-F1 | C-Index | t-AUC 1-yr | AUC | Macro-F1 |
| *Linear Modeling Approaches* | | | | | | | | |
| Lasso | 0.743 ±0.021 | 0.525 ±0.041 | 0.612 ±0.021 | 0.587 ±0.013 | 0.576 ±0.051 | 0.503 ±0.071 | 0.759 ±0.025 | 0.508 ±0.048 |
| Elastic Net | 0.724 ±0.015 | 0.495 ±0.036 | 0.661 ±0.009 | 0.548 ±0.025 | 0.634 ±0.049 | 0.520 ±0.083 | 0.768 ±0.022 | 0.517 ±0.034 |
| SVC | 0.766 ±0.030 | 0.537 ±0.038 | 0.712 ±0.010 | 0.551 ±0.011 | 0.650 ±0.043 | 0.513 ±0.069 | 0.787 ±0.010 | 0.530 ±0.032 |
| *Deep Learning Approaches* | | | | | | | | |
| MLP (3-layer) | 0.755 ±0.031 | 0.531 ±0.046 | 0.795 ±0.004 | 0.667 ±0.021 | 0.474 ±0.034 | 0.540 ±0.060 | 0.763 ±0.030 | 0.537 ±0.041 |
| MLP (5 layer) | 0.757 ±0.025 | 0.558 ±0.051 | 0.796 ±0.004 | 0.656 ±0.017 | 0.470 ±0.059 | 0.514 ±0.081 | 0.749 ±0.021 | 0.490 ±0.037 |
| *Node Classification Variants - Patient Similarity Network (Wang et al., 2021)* | | | | | | | | |
| MOGONET+Sage | 0.764 ±0.023 | 0.575 ±0.035 | 0.811 ±0.023 | 0.711 ±0.022 | 0.601 ±0.084 | 0.502 ±0.095 | 0.745 ±0.020 | 0.505 ±0.050 |
| MOGONET+GAT | 0.807 ±0.037 | 0.606 ±0.053 | 0.832 ±0.009 | 0.713 ±0.025 | 0.549 ±0.113 | 0.543 ±0.126 | 0.801 ±0.007 | 0.560 ±0.030 |
| MOGONET+GIN | 0.720 ±0.031 | 0.505 ±0.065 | 0.818 ±0.015 | 0.709 ±0.012 | 0.574 ±0.062 | 0.571 ±0.053 | 0.759 ±0.024 | 0.523 ±0.060 |
| *Graph Classification Variants - PPI Context Injection (Schulte-Sasse et al., 2021)* | | | | | | | | |
| EMOGI+Sage | 0.821 ±0.031 | 0.618 ±0.041 | 0.763 ±0.015 | 0.642 ±0.028 | 0.628 ±0.071 | 0.582 ±0.098 | 0.698 ±0.026 | 0.532 ±0.055 |
| EMOGI+GAT | 0.834 ±0.029 | 0.629 ±0.048 | 0.781 ±0.012 | 0.665 ±0.031 | 0.591 ±0.096 | 0.598 ±0.108 | 0.743 ±0.018 | 0.565 ±0.042 |
| EMOGI+GIN | 0.757 ±0.034 | 0.531 ±0.059 | 0.792 ±0.018 | 0.681 ±0.019 | 0.612 ±0.055 | 0.614 ±0.061 | 0.712 ±0.031 | 0.544 ±0.067 |
| *Biomolecule Instruction Tuning* | | | | | | | | |
| vicuna7bv1.5+MLP | 0.781 ±0.028 | 0.542 ±0.047 | 0.798 ±0.012 | 0.671 ±0.024 | 0.598 ±0.065 | 0.559 ±0.074 | 0.774 ±0.021 | 0.548 ±0.043 |
| vicuna7bv1.5+NODE | 0.815 ±0.032 | 0.601 ±0.039 | 0.827 ±0.015 | 0.718 ±0.021 | 0.612 ±0.078 | 0.575 ±0.089 | 0.798 ±0.018 | 0.571 ±0.038 |
| **KRONOS** | **0.857 ±0.025** | **0.673 ±0.031** | **0.849 ±0.011** | **0.742 ±0.018** | **0.664 ±0.058** | **0.628 ±0.067** | **0.823 ±0.014** | **0.618 ±0.029** |

Table 3: Performance comparison across different modeling approaches on CPTAC/TCGA outcomes. Best values per block are bolded, second best are underlined.

Additionally, we evaluate the optimal representation to be integrated into multi-modal LLM using three proteomics representation encoders: an MLP encoder processing raw features, a node encoder for patient similarity networks, and our proposed graph encoder (KRONOS) for PPI networks.

It is important to note that these predictive tasks are different from the instruction-following tasks. Thus, we perform an additional supervised fine-tuning step for KRONOS. A linear probe is added on top of KRONOS and trained for each prognostic predictive task.

The results on the prognostic benchmarks are found in Table 3, highlighting that KRONOS consistently surpasses all baseline models across the four predictive tasks. In summary, KRONOS exceeds baseline approaches, obtaining the highest performance in mortality prediction (AUC: 0.857, F1: 0.673), cancer type classification (AUC: 0.849, Macro-F1: 0.742), overall survival estimation (C-Index: 0.664, 1-yr t-AUC: 0.628), and disease stage prediction (AUC: 0.823, Macro-F1: 0.618).

The superior performance of graph-based approaches over linear methods highlights a fundamental limitation in proteomics analysis: proteomics signals that contribute to patient outcomes emerges from complex molecular interactions rather than individual protein abundance. Linear models like Lasso and ElasticNet assume protein features are independent of each other, failing to capture the intricate protein-protein dependencies that drives disease mechanisms. In contrast, KRONOS grounds representation learning in biological graphs to model these critical interactions, enabling the discovery of protein complexes that linear approaches cannot detect. This interaction-oriented modeling is crucial in cancer biology, where oncogenic processes often involve coordinated disruption of multiple interconnected proteins rather than isolated biomarkers.

Surprisingly, we found that MLPs without pre-aligned graph structure also performed competitively, suggesting that instruction-tuned language models can learn implicit signals from raw data. However, the explicit incorporation of PPI network structure in KRONOS still provides substantial improvements, validating that structured biological knowledge enhances clinical prediction capabilities.

## 5.2 ABLATION STUDIES

The ablation study in Table 4 compares optimal proteomics representations for semantic alignment into the LLM latent space through multiple graph and node encoders. The biomolecular instruction tuning framework reveals that graph encoders consistently outperform node encoders across

all tasks and GNN architectures. Among graph encoders, GAT achieves the best performance with mortality prediction (AUC: 0.857, F1: 0.673), cancer type classification (AUC: 0.849, Macro-F1: 0.742), overall survival (C-Index: 0.664, 1-yr t-AUC: 0.628), and stage classification (AUC: 0.823, Macro-F1: 0.618), followed by GIN and then GraphSAGE. The performance gap between graph and node encoders is substantial, with GAT-based graph encoders showing improvements of 4.2% AUC in mortality prediction, 2.2% AUC in cancer type classification, and 5.2% C-Index in survival prediction compared to their node encoder counterparts. This demonstrates that personalized PPI graph representations capture richer molecular interaction patterns than patient similarity networks when aligning representations to the semantic latent space, validating the core hypothesis that protein-protein interaction topology provides superior contextualization for proteomics data in precision medicine applications.

| Model | Mortality Pred. | | Cancer Type | | OS Prediction | | Stage Class. | |
|---|---|---|---|---|---|---|---|---|
| | AUC | F1 | AUC | Macro-F1 | C-Index | t-AUC 1-yr | AUC | Macro-F1 |
| *Biomolecular Instruction Tuning: Patient-specific PPI Graph Encoder* | | | | | | | | |
| Vicuna7bv1.5+Sage | 0.832 ±0.029 | 0.641 ±0.038 | 0.823 ±0.016 | 0.715 ±0.025 | 0.638 ±0.062 | 0.601 ±0.071 | 0.798 ±0.019 | 0.592 ±0.034 |
| **Vicuna7bv1.5+GAT** | **0.857 ±0.025** | **0.673 ±0.031** | **0.849 ±0.011** | **0.742 ±0.018** | **0.664 ±0.058** | **0.628 ±0.067** | **0.823 ±0.014** | **0.618 ±0.029** |
| Vicuna7bv1.5+GIN | 0.821 ±0.033 | 0.625 ±0.042 | 0.835 ±0.014 | 0.728 ±0.022 | 0.645 ±0.056 | 0.615 ±0.064 | 0.807 ±0.021 | 0.601 ±0.037 |
| *Biomolecular Instruction Tuning: Patient Similarity Node Encoder* | | | | | | | | |
| Vicuna7bv1.5+Sage | 0.798 ±0.035 | 0.578 ±0.043 | 0.815 ±0.017 | 0.706 ±0.023 | 0.601 ±0.072 | 0.562 ±0.081 | 0.785 ±0.020 | 0.559 ±0.041 |
| **Vicuna7bv1.5+GAT** | **0.815 ±0.032** | **0.601 ±0.039** | **0.827 ±0.015** | **0.718 ±0.021** | **0.612 ±0.078** | **0.575 ±0.089** | **0.798 ±0.018** | **0.571 ±0.038** |
| Vicuna7bv1.5+GIN | 0.787 ±0.038 | 0.565 ±0.045 | 0.821 ±0.018 | 0.712 ±0.025 | 0.595 ±0.069 | 0.558 ±0.077 | 0.779 ±0.022 | 0.553 ±0.043 |

Table 4: Performance comparison of Vicuna7bv1.5-based models on CPTAC/TCGA dataset. Best values per block are bolded, second best are underlined.

# 6 CONCLUSION

We present KRONOS, a novel graph-LLM architecture that grounds patient-specific proteomics in molecular interaction networks for clinical reasoning. Standard proteomics approaches lack semantic reasoning capabilities for complex clinical inference, while multi-modal LLMs cannot leverage protein-protein interaction network topology. KRONOS addresses these limitations by preserving molecular signature representation through interaction networks while enabling contextual prognostic reasoning via patient-centric instruction tuning.

While our proposed method demonstrates significant improvements in prognostic prediction of molecular signatures across the CPTAC cohort, several limitations warrant consideration for future development and clinical translation:

1. During inference and deployment, graph learning architectures are highly sensitive to distribution shifts. Further work needs to be done regarding the generalizeability of this architecture to other institutional datasets.

2. Graph construction requires substantial resources, and training both the LLM and encoder with our instruction tuning paradigm demands significant computational resources. This may restrict deployment in clinical environments, where resources may be limited. Further investigation must be done for translation into real-time diagnostic applications.

In summary, the superior performance of the graph representations for LLM integration compared to standard deep learning approaches for semantic alignment underscores the fundamental idea that rich modality representations yield improved prognostic reasoning and contextual understanding of patient-specific molecular signatures.

## ACKNOWLEDGEMENTS

We acknowledge the use of AI tools for assistance with manuscript writing, editing, and formatting. All scientific content, methodology, and results are original work by the authors.

REPRODUCIBILITY

To ensure reproducibility, we provide comprehensive implementation details and resources. Complete source code for KRONOS, including model architecture, training procedures, and evaluation scripts, is available at `https://anonymous.4open.science/r/src_biomolecular_instruction_tuning-1E0E/README.md`.All hyperparameters, training configurations, and experimental settings are specified in Appendix. The CPTAC-PROTSTRUCT instruction tuning dataset will be made publicly available upon publication. We used standard computational environments (Python 3.8, PyTorch 1.12) with specific package versions listed in the provided respository. Detailed preprocessing steps for CPTAC proteomics data, curated queryable proteins, and STRING PPI network construction are documented in the main text, along with inclusion in the repository. All experimental results can be reproduced using the provided code and data with the specified random seeds.

ETHICS

This study utilizes publicly available proteomics data from the Cancer Proteomics Tumor Analysis Consortium (CPTAC), which is accessible through the National Cancer Institute's Cancer Research Data Commons. All CPTAC data was collected under appropriate institutional review board (IRB) approval and patient consent for the original studies. Patient data has been de-identified in accordance with HIPAA guidelines. Our use of this publicly available dataset for computational analysis does not require additional IRB approval, as we do not have access to personally identifiable information and are conducting secondary analysis of previously collected, consented data. All analysis adheres to the data use agreements and access policies established by the National Cancer Institute.

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

# A  APPENDIX

| Model Type | Parameter | Search Space |
|---|---|---|
| SVC | C | [1e-4, 10] |
| | Gamma | {'scale', 'auto'} |
| | Kernel | {'linear', 'rbf', 'poly', 'sigmoid'} |
| | Degree | {2, 3, 4} |
| | Probability | {True, False} |
| Linear Models | C (Elastic-net, Lasso) | [1e-4, 10] |
| | L1-ratio (Elastic-net) | [0, 1.0] |
| | Max Iterations | {1000, 2000, 5000} |
| | Tolerance | [1e-5, 1e-4] |
| Deep Learning | Learning Rate | {1e-4, 1e-3, 1e-2} |
| | Dropout | {0.2, 0.3, 0.4, 0.6} |
| | Batch Size | {16, 32, 64, 128} |
| | Weight Decay | [1e-6, 1e-3] |
| | Epochs | {50, 100, 150, 200} |
| Graph Neural Networks | GNN Type | {'gin', 'gat', 'sage'} |
| | Hidden Dimensions | {64, 128, 256, 512} |
| | Number of Layers | {2, 3, 4} |
| | Learning Rate | {1e-4, 5e-4, 1e-3, 5e-3, 1e-2} |
| | Dropout | {0.3, 0.4, 0.5, 0.6} |
| | Weight Decay | [1e-5, 1e-2] |
| | Epochs | {100, 150, 200, 300} |
| | Batch Size (Graph Classification) | {8, 16, 32} |
| Patient Similarity GNN | K Neighbors | {5, 10, 15, 20} |
| PPI Network GNN | Pooling Strategy | {'mean', 'max'} |

Table 5: Hyperparameters and search space for baseline models.

| Parameter Category | MLP LLM | Node LLM | Graph LLM |
|---|---|---|---|
| Vision Tower Type | mlp | node_encoder | graph_tower |
| Architecture Type | mlp_3, mlp_5 | gcn, gat, sage, gin | gcn, gat, sage, gin |
| Hidden Size | 256, 512 | 512, 768, 1024 | 512, 768, 1024 |
| Dropout Rate | 0.1, 0.3, 0.5 | 0.1, 0.3, 0.5 | 0.1, 0.3, 0.5 |

Table 6: Model architecture search space for multi-modal LLM models.

| Parameter | MLP LLM | Node LLM | Graph LLM |
|---|---|---|---|
| Batch Size | 80, 100, 160 | 100, 120, 140 | 100, 120, 140 |
| Learning Rate | 2e-3, 3e-4, 1e-4 | 2e-3, 3e-4, 1e-4 | 2e-3, 3e-4, 1e-4 |
| Weight Decay | 0.01, 0.001 | 0.01, 0.001 | 0.01, 0.001 |
| Warmup Ratio | 0.03, 0.1 | 0.03, 0.1 | 0.03, 0.1 |
| Training Recipe | common, qlora_int8 | common, qlora_int8 | common, qlora_int8 |

Table 7: Training configuration search space for multi-modal LLM models.

| Parameter | MLP LLM | Node LLM | Graph LLM |
|---|---|---|---|
| **Number of Proteins** | 4792 (fixed) | 4792 (fixed) | Variable (graph-dependent) |
| **MLP Layers** | 3, 5 | N/A | N/A |
| **K-Neighbors** | N/A | 5, 7, 10, 15 | N/A |
| **GNN Layers** | N/A | 2, 3, 4 | 2, 3, 4 |
| **Attention Heads (GAT)** | N/A | 1, 4, 8 | 1, 4, 8 |
| **Graph Construction** | Direct features | Cosine similarity k-NN | Pre-built PPI graphs |
| **Pooling Strategy** | Single token | Node embedding | Global mean pooling |

Table 8: Model-specific parameters and configurations.

