# OpenReview forum: "Patient-Specific Biomolecular Instruction Tuning of Graph-LLMs"
_ICLR.cc/2026/Conference — ICLR 2026 Conference Desk Rejected Submission_

### Official Review · Reviewer_Dkd1 · 2025-10-25

**Soundness:** 4
**Presentation:** 3
**Contribution:** 3
**Rating:** 8
**Confidence:** 3

**Summary:**

In this work, the authors finetuned LLMs to accurately extract from soft tokens <1> precise and verifiable information about patient proteomics data (such as relative and absolute abundance values), but also <2> the answer to more ambitious questions about the patient (such as future treatment response, mortality predictions, and other clinical reasoning subtasks). These questions are generated synthetically using both ground truth data and synthetic augmentations by LLMs. The dataset will be released upon publication.

To encode the proteomics data before projection to soft tokens, a graph neural network is used. Multiple algorithms are considered, with their representative power evaluated by the downstream tasks. Approaches that do not rely on the graph itself are also evaluated. The approach of the authors, named KRONOS, outperforms linear models, patient-similarity graph baselines, and LLM+tabular encoders. The reported ablation studies suggest that graph encoders, especially GAT, outperform basic node encoders.

**Strengths:**

* The authors release (one of) the first patient-specific proteomics instruction-tuning corpus for oncology.
* The injection of a PPI-graph embedding as a special token into an LLM follows the recent trend initiated by VLMs, and applies it to the combination of structured biological topology and natural-language reasoning.
* The authors reported systematic ablations across multiple GNN families (SAGE/GAT/GIN) and a comparison against node encoders and MLP encoders within the same LLM backbone, giving ample confidence about their findings.
* Evaluation and syntetic data generation ensures that both hard-verifiable and more creative tasks are both represented, enabling cross-evaluation and rule out mere task adaptation without incorporation of real patient data in all their complexity.
* The usage of two stages (first hard data, then more open question answering) enables precise analysis of the model capabilities learned by the various contributions to the dataset, which is very valuable.

**Weaknesses:**

* The metrics on the basic synthetic tasks (e.g. relative and absolute abundances) are not reported. It would be interesting to know how the use-case finetunings affect the performance of the basic tasks, if at all, and how well the model performs on them.
* The usage of the STRING PPI network to create the graphs is never truly ablated. While the paper proves that the graphs created work better than an MLP per node and a node encoder, all these alternatives seem to share the same node structure, and it's not entirely clear to me how the feature representation of these nodes are computed.
* Using DeepSeek-R1-distilled models to generate the questions will probably not generate the diversity of questions hoped initially, as RL models are known for their lack of generative diversity. An improved version of the dataset would need to use several different models to generated the questions and paraphrases, as a proxy for a larger and more diverse downstream usage.
* The paper demonstrates that KRONOS beats LLM+MLP/NODE, but a strong graph-only predictor without an LLM is not reported in the same training data regime. This control would be needed to disentangle the contribution of instruction-tuned LLMs versus better graph representations.
* The parameter capacity and the added FLOPS required by the model types are not fully documented. Could it be possible that KRONOS simply benefits from larger effective capacity or more training compute than the non-graph LLM baselines? Was the performance of the network converged when the training ended, or were they all still improving?

**Questions:**

* Have you considered the impact of the number of soft tokens used to represent the patient, and its placement in the prompt? These factors are known to impact VLMs significantly.
* Did you consider other designs than STRING PPI networks for creating your graphs? Did you try alternative PPI/interaction graphs (BioGRID, IID, HINT, HuRI, Reactome FI) or integrate multiple sources? How sensitive are results to the graph choice and confidence cutoff?
* Did you consider weighting the nodes by prevalence - suprise in the MLP baslines, to separate the effect of simple attention over the effect of the graph connections (cross-attention)?
* Would you be able to provide the training time, GPU hours, and parameter counts of the considered models?
* Are there subgroup analyses (by cancer type, stage) showing where KRONOS helps most?

---

> ### Author Response · Authors · 2025-12-03
>
> We thank you for these very valuable comments, and we greatly appreciate your positive assessment and encouragement reflected in your score of “8.” We have considered your feedback and will revise the manuscript accordingly. Your insights have directly strengthened the clarity and rigor of the work, and we are grateful for your thoughtful suggestions. Below, we provide itemized responses to each of your comments.

---

> ### Author Response · Authors · 2025-12-03
>
> **[W2, Q2] Graph Construction, Node Features, and Ablations**
>
> *[W2]* We apologize for the lack of clarification in the computation of the feature representations. To thank the reviewer for giving us the opportunity to clarify our methods. The feature vector for protein $v$ in patient $i$ consists of the normalized $\log_2$ abundance  of that protein measured from mass-spectrometry assays. Raw intensities are  $\log_2$-transformed, z-scored within cancer type, and missing values are imputed using MICE,  yielding a complete node feature matrix with the same proteins across all patients. The difference between the MLP, node classification, and graph classification models are due to how these models learn representations from these identical features; MLP models treat the feature vector independently of any graph structure, node classification models construct a patient similarity network using these features as initial node embeddings to produce a node representation for each patient; and graph classification models propagate molecular information over the STRING PPI topology to generate a biologically contextualized graph representation for every patient. Importantly, we use the exact same node-feature matrix for all models (MLP, NODE, and GNN), ensuring that architectural comparisons isolate the effect of graph topology rather than differences in input representation.
>
> *[Q2]* We appreciate the reviewer’s suggestion and agree that benchmarking across multiple interaction resources is an important direction. In practice, although different studies have experimented with different topologies (e.g., BioGRID, HuRI), STRING is overwhelmingly the predominant choice in molecular GNN and biological graph representation research due to its broad proteome coverage, well-calibrated confidence scores, and unified single-relation structure. This makes STRING the standard for constructing molecular interaction graphs in prior work (e.g., EMOGI [1], GNN-SubNet [2], MTGCL [3], GCATSL [4]). Accordingly, we adopt STRING to maintain consistency with existing literature and cleanly isolate the contribution of our graph-LLM architecture and representation learning, rather than expanding the scope into a comparison of PPI databases.
>
> We agree that using alternative PPIs (BioGRID, IID, HINT, HuRI, Reactome FI) or fusing multiple resources into a single knowledge-base is valuable future work. We view the integration of these sources as a complementary research direction rather than the primary focus of the present paper. Accordingly, we will include this as a valuable future direction, and make the changes in our conclusion section accordingly.
>
> [1] EMOGI. Nature Machine Intelligence, 2022.
> [2] GNN-SubNet. Bioinformatics, 2023.
> [3]  MTGCL. IEEE Transactions of Computational Biology, 2023.
> [4] GCATSL. Bioinformatics, 2021.

---

> ### Author Response · Authors · 2025-12-03
>
> **[W4, Q3] Graph-Only Baselines, MLP Controls & Molecular Representation Learning.**
>
> *[W4]* We appreciate the reviewer’s concerns, and wanted to highlight the baseline models in our manuscript. Our manuscript includes the requested graph-only baselines trained under the same data and evaluation regime with no LLM backbone, where the representation itself is fed into the readout layer for classification. This includes MLP baselines, node classification paradigms with various node encoders, and graph classification variants with various graph encoders. As shown in Table 3, PPI graph classification variants substantially outperform both pure MLP and node-classification baselines, confirming that molecular network structure enriches predictive signals in the resulting representations. Notably, KRONOS still exceeds all graph-only models, indicating that the instruction-tuned LLM contributes additional gains beyond graph topology.
>
> *[Q3]* Furthermore, Table 4 shows ablations with various graph encoder towers for integration into KRONOS. We had a similar hypothesis that specific nodes contribute more biomedical signals towards downstream tasks, so we included Graph Attention Networks (GAT) as an encoder tower (GAT is also included in our graph representation learning variants without an LLM backbone). GAT explicitly learns attention coefficients over neighboring nodes, allowing the model to assign higher weights to proteins that are more informative within their local molecular context rather than relying on global prevalence or simple feature weighting. Graph-LLMs using GAT as the tower achieve the strongest performance, aligning with the choice of GAT as the final graph encoder for KRONOS in Table 3. We hope that this clarifies the reviewers concerns.

---

> ### Author Response · Authors · 2025-12-03
>
> **[W5, Q4, W3] Compute Budget, Convergence & Instruction Generation Diversity**
>
> *[Q4]* We thank the reviewer for the request, and want to provide the details for further transparency. We used 2xA6000 gpus for models without an LLM backbone (MLP, node classification, and graph classification variants with no LLM backbone). Additionally, to train KRONOS and its encoder tower variants, we utilized 2xH100 gpus. Stage 1 pretraining finished at ~4 hours, and Stage-2 instruction tuning finished ~1 or 2 hours. All models were trained with identical training criteria, with final model implementations included in the publicly available code. Additionally, we will include this information in our manuscript for further clarity.
>
> *[W5]* Regarding parameter capacity: we emphasize that KRONOS does not use label supervision during training, whereas mlp, node classification, and graph classification baselines with no LLM backbone are trained fully supervised on downstream labels with direct gradient signals. Thus, KRONOS does not benefit from additional task-specific capacity; rather, it benefits from unsupervised graph contextualization and alignment into the language space through multi-modal instruction tuning. As shown in Tables 3, graph-classification baselines with no LLM backbone already outperform multi-modal LLMs with MLP/NODE encoder towers under the same compute budget, indicating that the gains come from graph signal learning, not model size.
>
> In terms of convergence, all models, including KRONOS, reached stable loss curves. We did not observe continued improvements after the final checkpoints. We will clarify computation, parameter counts, and convergence behavior in the revised manuscript.
>
> *[W3]* We thank the reviewer for this thoughtful comment. We agree that using multiple generators can further increase linguistic diversity. Our choice of DeepSeek-R1-distilled was motivated by its strong reasoning capabilities and alignment quality, which we found important for producing clinically grounded, task-relevant instructions. While RL-based models may exhibit lower surface-level generative diversity, the resulting instruction set still spans a diverse range of clinical questions, reasoning tasks, and molecular interpretations, as reflected in the model’s performance across four distinct downstream tasks.
>
> Importantly, KRONOS achieves substantial gains despite relying on a single generator, suggesting that the method is robust to the level of diversity in the instruction source. We view incorporating multiple generators as a valuable extension to further broaden instruction coverage, and will clarify this in the manuscript and include it in our discussion of future dataset improvements.

---

> > ### Author Response · Authors · 2025-12-03
> >
> > **[W1, Q1] Synthetic Tasks, Soft Token Design in Language Modeling Objectives**
> >
> > *[W1]* The Stage-1 schema-alignment synthetic tasks were designed to ensure that the model can map patient-specific molecular graphs into the LLM’s semantic space before any downstream instruction tuning or reasoning exploration. In the current version, we did not re-evaluate these synthetic tasks after fine-tuning, as they were intended solely as pretraining scaffolds for representation alignment rather than as primary evaluation endpoints. We will clarify this intended role in the manuscript and note that reporting post-tuning performance is a valuable direction for future work.
> >
> > *[Q1]* We thank the reviewer for raising this point regarding multi-modal prompt injection into KRONOS. However, to clarify our architectural design, we don’t use soft prompts or a variable number of learned prompt tokens. Instead, we follow the multimodal architecture shown in Figure 1, where a single graph-derived embedding token is produced by the connector network and inserted at the beginning of the instruction sequence (Equation 4, page 6). This embedding functions as the multimodal representation of the patient’s proteomic profile and is processed by the LLM similarly to a vision token in LLaVA-style models. In our experiments, this design behaved stably, and we did not observe optimization issues that would necessitate exploring alternative placements.
> >
> > We agree that a systematic investigation of multi-token graph embeddings or alternative insertion positions could further enhance expressiveness, and we will note this as a valuable direction for future work.

---

### Official Review · Reviewer_vXGu · 2025-10-29

**Soundness:** 3
**Presentation:** 3
**Contribution:** 3
**Rating:** 6
**Confidence:** 3

**Summary:**

In this study, the authors proposed to integrate graph embedding with LLMs using instruction learning like in LLaVa, which achieved the highest performance in cancer mortality, type classification etc. It is a good improvement.

**Strengths:**

Using the instruction learning/turning to combine proteomics data graph with LLM for sample classification/prediction.
The evaluation results showed the best performance.

**Weaknesses:**

The prediction of cancer mortality, survival etc are not only based on omics data but also the other confounding factors, like age, gender, stage, ... which should be considered in the model.
In cancer biology, phenotype prediction (with the limited performance improvement) is less important than target and mechanism identification. It is unclear if the model can uncover the mechanisms associated with these phenotypes or not.

**Questions:**

It is unclear if the model can uncover the mechanisms associated with these phenotypes or not.
Adding the confounding factors in the prediction, in addition to the proteomics data.

---

> ### Author Response · Authors · 2025-11-24
>
> We thank you for these very valuable comments. We appreciate your encouragement of a “6”. We have carefully considered all of your input, and will revise our manuscript accordingly. We sincerely believe your input has helped us elevate our manuscript and we thank you for your suggestions. Below, please find itemized responses to your comments.
>
> **Integration of Confounding Factors:** We appreciate your biomedical insights and agree that clinical outcomes such as mortality, survival, and stage are influenced by molecular and non-molecular factors (e.g., age, sex, and clinical stage). We want to clarify that our evaluation intentionally isolates the prognostic signal derived from patient-specific proteomics to determine whether molecular data alone would be predictive of patient prognostics, a necessary step before constructing joint clinical–molecular models. As shown in the CPTAC dataset description (Section 3.1) and clinical QA generation (Section 3.4), demographic and stage-related metadata (age, sex, tumor stage) are available and can be incorporated into KRONOS via embedding concatenation or structured clinical tokens. However, introducing these covariates would merge molecular and non-molecular signals, making it difficult to attribute gains to the graph-LLM architecture itself rather than to known clinical risk factors. Nevertheless, integrating clinical information into KRONOS is an impactful and high-priority extension for increasing clinical translatability, and we will explicitly add this direction to the revised manuscript.
>
> **Clarifications on mechanistic discovery:** We appreciate your inquiry about interpretability and mechanistic discovery. We appreciate the opportunity to clarify aspects of our architecture. KRONOS does not claim to perform explicit mechanistic identification (e.g., causal pathway discovery or validated target inference). However, its architecture is designed to enable such analyses: our original intention was to indicate that the graph-based LLM architecture provides a biological grounding towards interpretability through the explicit encoding of protein–protein interaction topology, rather than to claim that the present work includes empirical interpretability evaluations. The proposed architecture focuses on establishing predictive validity of aligning PPI-grounded representations towards the LLM language space for improved patient prognosis. We agree that systematic attention-based or saliency-based mechanistic analysis would be highly impactful for clinical trustworthiness and transparency, and we will emphasize this as a major direction for future work.

---

### Official Review · Reviewer_V6MQ · 2025-10-31

**Soundness:** 2
**Presentation:** 2
**Contribution:** 2
**Rating:** 6
**Confidence:** 2

**Summary:**

This paper introduces CPTAC-PROTSTRUCT, the first large-scale instruction tuning dataset for proteomic understanding in oncology, and proposes KRONOS, a graph-LLM architecture that integrates protein-protein interaction (PPI) networks with patient-specific proteomics for clinical reasoning tasks. CPTAC-PROTSTRUCT contains over 370k examples derived from the CPTAC cohort, targeting schema alignment and clinical reasoning challenges unique to the proteomics domain. KRONOS augments language modeling capability with structured graph representations of patient omics, enabling improved performance on benchmark clinical tasks, including mortality prediction, cancer type, survival, and staging.

**Strengths:**

1. CPTAC-PROTSTRUCT is a large and thoughtfully curated dataset, tackling both schema-level alignment and clinical reasoning, and offers clear value as a resource for future work.

2. Experimental results (Table 3) demonstrate that KRONOS outperforms multiple strong baselines, including linear models, MLPs, and both node- and graph-based approaches in a rigorous 5-fold cross-validation. For example, KRONOS achieves AUC 0.857 (±0.025) and Macro-F1 0.742 on key tasks, emphasizing tangible improvements over prior state-of-the-art.

**Weaknesses:**

1. The paper claims improved interpretability via the use of graph-based molecular knowledge, but lacks a concrete illustration of this point. For instance, no case studies or saliency analyses are provided showing which proteins/interactions the model relies on for clinical reasoning. Counterexamples or failure analyses are absent as well.

2. The authors only compare against classical and GNN models. There is no comparison to recent instruction-tuned LLMs applied to medical or omics data (e.g., BioInstruct, scGPT), workflows that could be strong or even more competitive baselines; this makes claims of state-of-the-art incomplete.

3. The rationale for freezing the LLM backbone (Section 4.2.1) in schema alignment and then fine-tuning in clinical reasoning is not well-motivated. Do alternatives (e.g., full fine-tuning vs. partial) yield better/worse results? No ablation or discussion exists.

**Questions:**

1. Please clarify the architectural and training specifics for the connector network and GNN encoder, including hyperparameters and merging strategy. For instance, what is the dimensionality of $\mathbf{z}_i$ and $\mathbf{e}_i$, and is the pooling strategy (max, mean, attention) critical?

2. Is the two-stage curriculum training approach significantly better than joint or single-stage training? Any quantitative or qualitative support?

3. Can you extend the analysis by providing interpretability visualizations (e.g., attention maps, gradient-based saliency) to demonstrate how the model reasons over proteins/interactions during clinical tasks?

4. Could you elaborate on procedures used to strictly segment patient-level data between instruction datasets, schema alignment, and benchmarking? How is the fold assignment managed when multiple template-derived questions per patient exist?

---

> ### Author Response · Authors · 2025-11-24
>
> We thank you for these very valuable comments. We appreciate your encouragement of a “6”. We have carefully considered all of your input, and will revise our manuscript accordingly. We sincerely believe your input has helped us elevate our manuscript and we thank you for your suggestions. Below, please find itemized responses to your comments.
>
> **[W2, W1, Q3] Clarification on Comparison to Instruction-Tuned LLMs and Interpretability Claims:**
>
> *[W2]* We thank the reviewer for highlighting this point of comparison against existing instruction-tuned LLMs. We agree that comparing performance against recent instruction-tuned LLMs would be valuable where feasible. However, existing instruction-tuned biomedical models (e.g., BioInstruct [1], scGPT [2]) are not directly comparable to KRONOS because they lack the ability to process patient-level proteomics values or personalized PPI graphs. BioInstruct was designed to perform text-based clinical NLP tasks, and scGPT models single-cell gene-expression sequences by assuming cell-level tokenization; both of these models do not support patient-specific omics-informed molecular networks or multi-modal alignment of proteomics data. KRONOS addresses this distinct challenge by integrating individualized proteomics signatures with PPI topology and aligning these structured molecular representations with LLM through multi-stage instruction-tuning. No existing model architecture to our knowledge provides a mechanism to reason across proteomics assays for patient endpoints, or to represent patient-specific omics-informed molecular networks in the language space. Due to these factors, a direct empirical comparison is methodologically incompatible. Instead, we benchmark against the strongest models that accept the same modality against the same evaluation paradigm, classical ML, MLPs, patient similarity network GNNs, and PPI-GNNs, and demonstrate consistent improvements from incorporating PPI-grounded representations into an instruction-tuned LLM framework. This information will be clarified in our manuscript, upon acceptance of a camera-ready submission.
> [1] Hieu et al. BioInstruct. arXiv preprint arXiv:2310.19975, 2023.
> [2] Haotian et al. scGPT. Nature Methods, 21(8):1470–1480, 2024.
>
> *[W1, Q3]* We thank the reviewer for this helpful observation of our claim in interpretability. The original phrasing may have overstated the extent of our analysis. Our original intention was to indicate that the graph-based LLM architecture provides a mechanistic direction towards interpretability through the explicit encoding of protein–protein interaction topology, rather than to claim that the present work includes empirical interpretability evaluations and shall revise the manuscript to clarify this distinction. We emphasize that the primary contributions of this work are (1) introducing the first proteomics-focused instruction-tuning dataset and (2) demonstrating that aligning patient-specific omics-informed molecular graph representations through multi-modal instruction tuning improves clinical prognostic performance. Interpretability analyses such as attention-based explainability or saliency-based mapping over molecular nodes represents a valuable direction for future work, and will explicitly incorporate this in the revised manuscript to better articulate this scope.

---

> ### Author Response · Authors · 2025-11-24
>
> **[W3, Q1, Q2] Clarifications for Curriculum Tuning and Hyperparameters:**
>
> *[W3, Q2]* We thank the reviewer for these questions on curriculum training, and want to ensure clarity on our motivations of conducting multi-stage training. Our two-stage curriculum design follows established paradigms in multi-modal instruction tuning [1–5], which demonstrated that separating modality alignment from task-specific fine-tuning yields more stable representation learning and improved downstream performance. In Stage 1 training (schema alignment), we train the connector and GNN encoder to project proteomics graph representations into the LLM's pre-trained language space while freezing the LLM backbone to prevent catastrophic forgetting of pre-trained information, a strategy adopted in existing multi-modal LLM architectures, notably LLaVA [1], LLaVA-Med [2], MIMIC-Inst [5],  BLIP-2 [3], and MiniGPT-4 [4]. In Stage 2 (clinical reasoning), we unfreeze the LLM to enable joint adaptation of both the molecular encoder and language model for prognostic tasks, consistent with clinical instruction tuning approaches such as MIMIC-Instr [5] and other multi-modal LLMs referenced above [1, 2, 4]. We acknowledge that we did not include ablations comparing alternative training strategies (e.g., full fine-tuning from Stage 1, single-stage joint training). We will add discussion of this limitation in the revised manuscript; however, we emphasize that our primary contributions demonstrate that integrating PPI-grounded graph representations into an LLM improves clinical prediction performance, and we adopted a well-validated training protocol to ensure stable multi-modal alignment. Accordingly, we will clarify this information in our manuscript.
>
> [1] Liu et al. Visual instruction tuning. NeurIPS 2023.
> [2] Li et al. LLaVA-Med. NeurIPS 2023.
> [3] Li et al. BLIP-2. ICML 2023.
> [4] Zhu et al. MiniGPT-4. ICLR 2024.
> [5] Wang et al. MIMIC-Instr. NeurIPS 2024.
>
> *[Q1]*  We apologize for the lack of transparency of finalized hyperparameters used during multi-modal instruction tuning. Accordingly, below is a comprehensive summary of KRONOS' architectural and training specifications, including GNN encoder configuration, connector network design, and stage-specific training parameters. In addition, full end-to-end training code for KRONOS and a README.md is attached in the manuscript for full transparency and reproducibility. We will ensure that the following table is explicitly mentioned for complete clarity. Regarding merging, the processed molecular graph embedding $e_i$ is integrated into the instruction sequence as a specialized token. We prepend the graph embedding to the tokenized text input, forming a multi-modal sequence $T_{\text{multi}} = [e_i, t_1, \ldots, t_n]$ that is then processed by the LLM, as established in current multi-modal LLM literature [1-5] (cited above).

---

> ### Author Response · Authors · 2025-11-24
>
> *[Q1 Continued]*
> | **Component**   | **Parameter**       | **Stage 1 (Schema Alignment)** | **Stage 2 (Clinical Reasoning)** |
> | --------------- | ------------------- | ------------------------------ | -------------------------------- |
> | **GNN Encoder** | Type                | GAT                            | GAT                              |
> |                 | Hidden Dim  | 512                            | 512                              |
> |                 | Dropout             | 0.3                            | 0.3                              |
> |                 | Pooling             | Global Mean                    | Global Mean                      |
> |                 | Tuning              | Full                           | Full                           |
> | **Connector**   | Architecture        | 2-layer MLP + GELU             | 2-layer MLP + GELU               |
> |                 | Output Dim (*d*) | 4096                           | 4096                             |
> |                 | Tuning              | Full                           | Full                             |
> | **LLM**         | Model               | Vicuna-7B-v1.5                 | Vicuna-7B-v1.5                   |
> |                 | Tuning              | Frozen                         | QLoRA (8-bit)                    |
> |                 | LoRA Rank (*r*)     | —                              | 64                               |
> |                 | LoRA Alpha (α)      | —                              | 128                              |
> |                 | LoRA Dropout        | —                              | 0.1                              |
> | **Training**    | Learning Rate       | 2e-3                           | 3e-4                             |
> |                 | Batch Size          | 100                            | 100                              |
> |                 | Epochs              | 1                              | 3                                |
> |                 | Warmup Ratio        | 0.1                            | 0.03                             |
> |                 | Weight Decay        | 0.01                           | 0.01                             |
> |                 | Optimizer           | AdamW (β₁=0.9, β₂=0.95)        | AdamW (β₁=0.9, β₂=0.95)          |
> |                 | LR Scheduler        | Cosine                         | Cosine                           |
> |                 | Precision           | FP16 + TF32                    | FP16 + TF32                      |

---

> ### Author Response · Authors · 2025-11-24
>
> **[Q4] Data Segmentation and Fold Assignment:**
>
> We thank the reviewer for this important question regarding data segmentation and leakage prevention. All data splits are performed at the patient level, ensuring that every schema-alignment or clinical-reasoning QA pair derived from a given patient is assigned to the same fold. This guarantees that no molecular or instruction-derived information from a patient in the test set appears in the training set.
>
> CPTAC-PROTSTRUCT contains two subsets: schema-alignment QA pairs (Stage 1), and 26,157 clinical-reasoning QA pairs (Stage 2). However, both subsets originate from the same cohort of 1,658 patient samples. Although multiple template-generated questions may be associated with each patient, the patient-to-fold mapping is fixed and shared across both subsets.
>
> For prognostic prediction (Table 3), we perform 5-fold nested cross-validation at the patient level using stratified sampling to maintain cancer-type balance across folds. If a patient is assigned to a specific fold, all associated instruction pairs linked to the patient are assigned to that fold, and during evaluation, the model never observes any schema or reasoning-stage questions associated with those patients.
>
> The random seeds, fold assignments, and full dataset will be released upon acceptance to ensure full reproducibility. We will add an explicit description of this patient-level segmentation procedure in the revised manuscript.

---

### Official Review · Reviewer_1ZLW · 2025-11-01

**Soundness:** 2
**Presentation:** 2
**Contribution:** 2
**Rating:** 4
**Confidence:** 4

**Summary:**

This paper introduces both a dataset, CPTAC-PROTSTRUCT, and a model for analyzing it, called KRONOS.  CPTAC-PROTSTRUCT  is an instruction-tuning corpus that pairs patient-specific proteomic profiles from CPTAC with schema-alignment prompts and clinically framed question-answer pairs. The authors curate proteins using centrality, variance, and cancer knowledge bases, then generate 354k paraphrased schema questions and 26k clinical reasoning items to adapt a general LLM to the proteomics modality.
The companion model, KRONOS, embeds each patient’s proteomic measurements on a STRING PPI graph, encodes it with a GNN, maps the graph embedding into the LLM token space via a connector, and uses a two-stage curriculum for schema alignment and clinical reasoning.

On CPTAC-derived prognostic tasks, KRONOS consistently outperforms strong linear, MLP, patient-similarity, and graph baselines across multiple clinically relevant endpoints. The work’s primary contributions are: introducing a large, task-targeted instruction set for proteomics; proposing a clear graph-to-LLM integration via a learnable connector with a staged training curriculum; showing benefits over node-only and non-graph alternatives; and providing detailed preprocessing, code, and hyperparameter settings to facilitate reproducibility.

**Strengths:**

[S1] Overall workflow clarity:
The overall workflow figures are clear and easy to follow.

[S2] Dataset contribution:
 The instruction-tuning dataset is large and explicitly decomposed into schema alignment and clinical reasoning stages, aligning the curriculum with the model architecture. I believe this dataset is the paper’s biggest contribution.

[S3] Comparative study:
A comparative study spans multiple families of baselines, including linear models, MLPs, patient-similarity networks, and PPI-graph classifiers, facilitating a comparison of gains.

[S4] Consistent improvements:
Reported gains are consistent across four clinically meaningful endpoints, not just a single metric or task, which strengthens the case for this LLM-graph integration to be adopted by others.

**Weaknesses:**

[W1] Need for more detailed methodological descriptions:
While the authors have provided a mathematical description of KRONOS, it would be beneficial to include a companion figure that illustrates the model's inner workings. This visual aid would complement the text and provide a clearer understanding of the model. Currently, the diagrams that describe the pipeline are superficial, and more detail is needed regarding the inner workings of the model to facilitate understanding and reproducibility. Additionally, it would be helpful to more explicitly highlight the aspects of the model that differ from existing approaches that combine LLMs with graph-based architectures, so that the contribution is clearer to readers.

[W2] Phi operator definition: The definition for the $\phi$ operator related to the connector was not made explicit; the authors only refer to it as a `network`. Additionally, the relationship between W, b, and the connector should be stated in the main text. Judging from the code on the `tinyllava\modelconnector`, I can see that it can take multiple forms, but these need to be described thoroughly in the main text. The paper should be self-explanatory without requiring reference to the code for definitions.

[W3] Study generalizability: While it is acknowledged that the model is trained and evaluated on CPTAC, it is also important to test these limitations beyond CPTAC and TCGA, since they have a certain degree of overlap. To enhance the generalizability of this research, I recommend testing the model on an independent cohort or across institutions. This will help to probe for distribution shift and extend this work to other clinical settings.



[W4] Figure captions need to be improved: The figure captions could be improved, especially for figures 1 and 2. They should describe the content of the images and explain their significance in relation to the overall workflow.

**Questions:**

[Q1] Out-of-sample validation:
Regarding external validity and W3: Could the authors perform an out-of-sample evaluation using a non-CPTAC/TCGA cohort (since there is some overlapping data between these datasets)?

[Q2] Leakage controls: How do the authors guarantee that instruction generation, imputation, and graph construction are performed strictly within each training fold, with no statistics/calculations crossing splits? A simple per-fold data flow schematic or description would be helpful.

[Q3] Related to W1 and W4: Please define undefined terms, such as the connector, and provide more detailed descriptions of the model. This includes improving Figures 1 and 2 to reflect this change and making the framework more transparent.

---

> ### Author Response · Authors · 2025-11-24
>
> Thank you for these extremely valuable comments; we have carefully considered all of your input, and will revise our manuscript accordingly. We sincerely believe your input will help us enhance our manuscript and we thank you for your suggestions. Below, please find itemized responses to your comments.
>
> **[W1, W2, W4, Q3] Need for More Detailed Methodological Descriptions/Architectural Innovations:**
>
> *[W1, W4, Q3]* Thank you for this thoughtful comment. We agree this manuscript can be elevated through improved clarity of the overall architecture and will substantially revise the manuscript to address this. In the updated version, we will add a new figure that illustrates the full internal workflow of KRONOS to a substantial granularity, including (1) thorough construction of the patient-specific PPI graphs, (2) GNN neighborhood aggregation to enriched node embeddings, (3) transformation of these node embeddings to a graph embedding through a graph-level READOUT aggregation, (4) connector transformation that maps this patient-specific PPI graph embedding into the LLM token space, and (5) integration of this multi-modal token into the LLM during instruction tuning. This additional figure will reflect the text provided in the mathematical formulation and make KRONOS’ model operations transparent. We will also revise the captions for Figures 1 and 2 to provide more detailed descriptions of the visual content and explicitly connect each component to KRONOS' overall workflow. We thank the reviewer for this suggestion.
>
> Additionally, we will revise the text to explicitly highlight the key aspects of the model that differentiate KRONOS from existing graph-LLM approaches:  (1) the use of patient-specific omics-enriched PPI graphs to generate enhanced representations tailored to each individual patient, rather than generic molecular networks; (2) a two-stage proteomics-grounded instruction-tuning curriculum which is fundamentally different from prior multimodal LLM pipelines that do not provide modality-aware alignment or proteomics-specific supervision; and (3) the direct integration of patient-specific molecular graph representations into the LLM token space through a connector explicitly grounded in omics-enriched PPI topology, which differ to the current scope of graph-LLMs that operate on generic graph embeddings without patient-level molecular context or biological interaction structure.
>
> *[W2]* Thank you for pointing out issues where the mathematical formulation could be clarified. We agree that the $\phi_{\text{connector}}$ and the utilized parameters should be more explicit in the main text. In the revised manuscript, we will add a precise mathematical definition of the connector as a learnable transformation
> $\phi_{\text{connector}}(z_i) = W z_i + b$ where $z_i$ is the patient-specific omics-informed graph embedding produced by the GNN, and $W$ and $b$ are trainable parameters that project this embedding into the LLM token-embedding space, following architecture design of established multi-modal LLMs, notably LLaVA [1], LLaVA-Med [2], MIMIC-Inst [3]. We will clarify that this formulation corresponds to the MLP connector used in KRONOS in the main text.
>
> Accordingly, all of the above information will be updated in the respective sections of the revised manuscript.
>
> [1] Liu et al. Visual instruction tuning. NeurIPS 2023. [2] Li et al. LLaVA-Med. NeurIPS 2023. [3] Wang et al. MIMIC-Instr. NeurIPS 2024.

---

> ### Author Response · Authors · 2025-11-24
>
> **[W3, Q1, Q2] Study Generalizability, Validation, and Leakage Mitigation:**
>
> *[Q1, Q2]* Thank you for raising this important point regarding external validity and generalization beyond CPTAC/TCGA. We completely agree that evaluating KRONOS under distribution shift is valuable. While we are currently acquiring and harmonizing an independent non-CPTAC proteomics cohort (ICPC datasets) and will update you on our performance as soon as possible, our current evaluation includes a 5-fold cross-validation setup with patient-level splits in which each fold’s test set is held out entirely from all leakage-sensitive stages, as graph construction is fully patient-specific (one graph per patient) and instruction pairs are generated exclusively from each patient’s own proteomic values. Protein filtering and MICE imputation are performed once within each cancer type, which is standard practice to maintain a consistent feature space and ensure stable imputation, but these steps only determine which proteins are retained and estimate missing values using cohort-level statistics. These steps do not use any information from the patient’s outcomes or from other fold-specific patterns, and they do not create patient-to-patient information mixing beyond cancer-type-level variance structure. All downstream steps that rely on per-patient values are strictly computed on a per-patient basis and do not mix information across folds. We will clarify this in our revised manuscript for full transparency and reproducibility.
>
> *[Q2, W3]* Additionally, we are actively exploring publicly available proteomics datasets with patient-level profiles and clinical endpoints, and have identified the ICPC datasets as a candidate for external validation, and will update this thread with the results accordingly. Nevertheless, we believe the existing 5-fold CV design, combined with our leakage-prevention protocol, provides meaningful evidence of KRONOS’s robustness, and we are committed to extending this assessment where feasible.
>
> Accordingly, all of the above information will be updated in the revised manuscript.

---

### Note · Program_Chairs · 2026-01-17
**Submission Desk Rejected by Program Chairs**

The following references in this submission do not refer to real documents and/or have major errors in bibliographic information:

     Zihao Wang, Xiaohan Yang, Hanbin Liu, et al. Mimic-instr: A large-scale instruction-following dataset for electronic health records. Advances in Neural Information Processing Systems, 2024.